# Deterministic all-optical magnetization writing facilitated by non-local transfer of spin angular momentum

Youri L. W. van Hees [1✉], Paul van de Meugheuvel [1], Bert Koopmans [1] & Reinoud Lavrijsen [1]

Ever since its discovery around a decade ago, all-optical magnetization switching (AOS) using femtosecond laser pulses has shown potential for future data storage and logic devices. In particular, single pulse helicity independent AOS in certain ferrimagnetic alloys and multilayers is highly efficient and ultrafast. However, in most cases it is a toggle mechanism, which is not desirable for applications. Here we experimentally demonstrate conversion from toggle switching to a deterministic mechanism by biasing AOS in a Co/Gd bilayer with a spin polarized current which is optically generated in an adjacent ferromagnetic reference layer. We show deterministic writing of an 'up' and 'down' state using a sequence of one or two pulses, respectively, and demonstrate the non-local origin by varying the magnitude of the generated spin current. Our demonstration of deterministic magnetization writing could provide an essential step towards the implementation of future optically addressable spintronic memory devices.

---

[1] Department of Applied Physics, Institute for Photonic Integration, Eindhoven University of Technology, P.O. Box 513, 5600 MB Eindhoven, the Netherlands. ✉email: y.l.w.v.hees@tue.nl

The explosive growth of data production and consumption rates in the past decades has driven the search for faster and more energy-efficient methods to record data. Among these methods, the use of optics to assist or even facilitate data recording in magnetic materials shows promise in terms of speed and energy efficiency[1]. More specifically, all-optical switching (AOS) of magnetic materials, whereby the magnetization can be reversed on a picosecond timescale using only femtosecond (fs) laser pulses, has striking potential. First discovered around a decade ago[2], it has since been shown that two mechanisms can be distinguished, namely, (1) multiple pulse helicity-dependent switching and (2) single pulse helicity-independent switching. The helicity-dependent mechanism has been observed in several magnetic materials[2–8], and is believed to result from a dependence of the absorption of circularly polarized light on the magnetization direction[9]. Although this mechanism is deterministic in that the final magnetization direction is defined by the helicity of the incident light alone, it requires multiple laser pulses[7], which limits speed and applicability. The second effect, single pulse helicity-independent switching, has thus far been demonstrated in ferrimagnetic GdFeCo alloys[10–12], synthetic ferrimagnetic Co/Gd bilayers[13] and very recently in a ferrimagnetic Heusler alloy[14]. This effect relies strongly on transfer of angular momentum between magnetic sublattices, as well as a difference in demagnetization timescales between the involved materials.

In the example of a ferrimagnetic rare earth-transition metal (GdFeCo) alloy, single pulse switching has been shown to be a toggle process[10]. In the ground state of these materials, the sublattices (Gd and FeCo) are aligned antiparallel due to an antiferromagnetic coupling. Upon fs laser pulse induced heating both sublattices will demagnetize, but FeCo will do so more rapidly than Gd. This means that at some point the FeCo magnetization will be nearly quenched whilst there is still a significant amount of Gd magnetization. Due to transfer of angular momentum between the sublattices, the FeCo magnetization is pulled through zero, creating a temporary ferromagnetic state. While the FeCo sublattice now remagnetizes in the opposite direction, Gd continues to demagnetize, and is also pulled through zero due to the antiferromagnetic coupling between the sublattices. After relaxation both sublattice magnetizations end up opposite to the initial state. It should be noted that single pulse helicity-dependent switching has also been observed in a narrow range of laser fluences for GdFeCo alloys[2,15], owing to magnetization- and helicity-dependent light absorption[16]. This same switching process has also been demonstrated in synthetic ferrimagnetic Co/Gd bilayers[13], which we will use in this work. Switching in the latter materials has been shown to be more robust than in ferrimagnetic alloys, in the sense that it does not depend on the sublattices being close to magnetization compensation[17].

Using a toggle mechanism for data storage applications would require prior knowledge of the state of a bit to overwrite it, imposing limits on speed and integration flexibility. Therefore it is desirable to find a deterministic AOS procedure using as few laser pulses as possible where the final magnetization direction is not always the opposite of the initial state but instead relies on a specific process to set and reset a magnetic bit. In this work, we will propose and experimentally demonstrate such a method.

Regarding the underlying physics, the single pulse AOS scenario described earlier has been confirmed by several complementary theoretical models[11,18–21]. These works have all included the exchange of angular momentum between the two sublattices as an essential ingredient to find switching. However, it is not yet clear to which extent this exchange is driven by local exchange scattering processes, or by non-local transfer of angular momentum. Recent work has shown that upon switching, angular momentum can be transferred from a switching layer to a ferromagnetic layer separated by a conducting spacer, thereby switching the ferromagnetic layer[22]. The inverse effect where angular momentum is transferred non-locally from a ferromagnetic reference layer to a switching layer has however not been addressed so far.

In this work, we experimentally demonstrate deterministic magnetization writing using either one or two laser pulses in a system consisting of a ferromagnetic *reference layer*, a conductive spacer layer, and an all-optically switchable *free layer*. When exciting the sample with an fs laser pulse, a spin-polarized electron current will be generated in the ferromagnetic reference layer, governed by its magnetization[23,24]. Angular momentum carried by this spin current is transferred to the free layer, where it can assist or hinder switching depending on the relative magnetization orientation of the reference and free layer. This asymmetry between the parallel and antiparallel alignment of these layers leads to two incident laser fluence regimes. Above a certain threshold fluence, the final state is completely determined by the orientation of the reference layer. When increasing the fluence above a higher threshold, the familiar toggle switching mechanism is recovered. We demonstrate experimentally how these two regimes combined can be used to deterministically write both magnetization states of the free layer, regardless of its initial state. Moreover, we confirm that this effect scales as expected with the optically generated spin current, and demonstrate that its magnitude can be easily tuned.

## Results

**Deterministic optical magnetization writing.** Our system consists of two magnetic layers, the previously mentioned reference and free layers. First is a ferromagnetic $Co(0.2 nm)/Ni(x))_{x4}/Co(0.2)$ multilayer, which acts as the reference layer. This layer has an out-of-plane magnetization due to strong perpendicular magnetic anisotropy (PMA) and a relatively large magnetic moment compared to the switching free layer. The free layer is composed of a synthetic ferrimagnetic Co/Gd bilayer (1 and 3 nm respectively) with PMA, which is known to exhibit toggle AOS[13]. It should be noted that due to the antiferromagnetic exchange between Co and Gd[25], the proximity-induced Gd magnetization (corresponding to ~0.5 nm at room temperature) is aligned antiparallel with the adjacent Co in the ground state. When discussing the state of the free layer, we will refer to the dominant compound, i.e. Co. The reference layer and free layer are separated by a 5-nm Cu spacer layer, which decouples the layers magnetically whilst being transparent for spin-polarized electrons[26]. The stack is deposited using DC magnetron sputtering on an Si/SiO2 substrate. More information on both the fabrication process and the full sample composition is available in the 'Methods' section.

All four magnetization states of the system (Co/Ni up/down and Co parallel/antiparallel) can be realized using an external magnetic field. This is illustrated in Fig. 1a using the out-of-plane hysteresis loop of the sample measured using the polar magneto-optical Kerr effect. Positive (+) and negative (−) states are defined as the reference layer having magnetization pointed up or down respectively, while parallel (P) and antiparallel (AP) refers to the relative orientation of the reference layer and Co in the free layer. The four states are thus defined as $P^+$, $AP^+$, $AP^-$ and $P^-$. Note that since corresponding + and − states are simply time reversed versions of each other, they do not add any new physics. Therefore, without loss of generality, we only consider the positive (+) P and AP states, and drop the plus sign in the remainder of this work. Data on the corresponding negative states are shown in Supplementary Fig. 1.

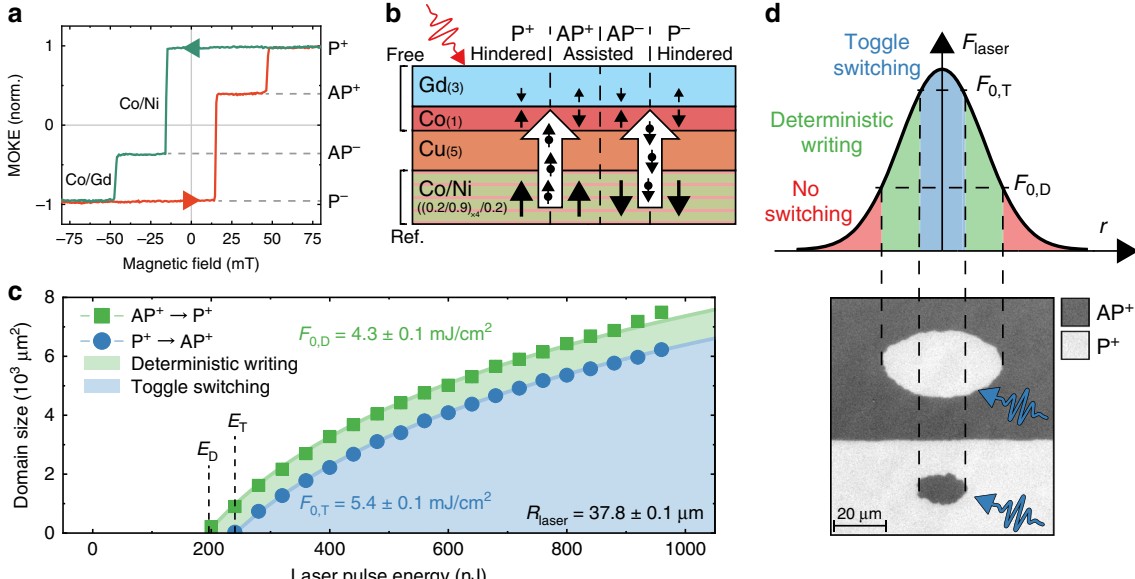

**Fig. 1 Demonstration of deterministic single pulse all-optical magnetization writing. a** Out-of-plane hysteresis loop of the $(Co/Ni)_{x4}/Co/Cu/Co/Gd$ sample used in this work. The Co/Ni multilayer and Co/Gd bilayer switch independently at different applied magnetic fields, resulting in four possible magnetization states. **b** Simplified sketch of the multilayer stack used in this work and how it breaks the toggle switching symmetry. The spin current generated in the reference layer upon demagnetization flows through the Cu spacer and aids or hinders switching in the free layer depending on the relative magnetization orientation of these layers. Numbers in parentheses indicate layer thicknesses in nm. **c** Measurement of optically switched domain size as a function of incoming laser pulse energy for a sample prepared in either $AP^+$ or $P^+$ state. Fitting is done assuming a Gaussian laser spot profile to extract the threshold laser fluence $F_0$. **d** Top: Sketch of the threshold fluences $F_{0,D}$ and $F_{0,T}$ relative to the Gaussian laser spot profile. Bottom: Kerr microscopy images of switched domains after excitation by a single laser pulse of two regions with different initial magnetization.

In our system both states of the free layer are no longer equivalent, due to a symmetry breaking provided by the reference layer. The nature of this symmetry breaking is sketched in Fig. 1b. Upon fs laser excitation, the ferromagnetic reference layer will demagnetize on a sub-picosecond timescale. The lost angular momentum is partially converted into a spin current, mediated by mobile electrons. Although multiple mechanisms for the generation of this spin current have been proposed, all find that the spin carried by the mobile electrons is aligned with the layer from which they originate[23,24,27], i.e. the reference layer. The resulting spin current flows through the conducting Cu spacer layer to the free layer, where the spin angular momentum is deposited via scattering between mobile and localized electrons. Such non-local transfer of spin angular momentum can have a measurable impact on the ultrafast magnetization dynamics of a magnetic layer[28–30]. Moreover, the spin current is expected to exist roughly on the timescale of the demagnetization of the reference layer, thereby transferring angular momentum to the free layer while Co and Gd are still demagnetizing. Because at this point the free layer is out of equilibrium with a strongly reduced magnetic moment, the additional angular momentum could significantly influence the switching process. We note that a spin current is also generated in the free layer, travelling towards the reference layer. However, as this particular spin current is much weaker due to the reduced magnetic moment of the free layer and is not found to significantly influence the receiving reference layer, we have not sketched it here. Moreover, it is precisely the threshold nature of AOS that allows us to observe the effect of a spin current much more clearly than in the reference layer, which is only partially demagnetized.

When the reference layer magnetization, and thereby the spin current polarization, is antiparallel to the Co magnetization in the free layer, the additional angular momentum should assist the demagnetization of Co and hinder the demagnetization of Gd. As switching is strongly dependent on the formation of a temporary

ferromagnetic state aligned with Gd[10], and this state can now form more easily, switching is assisted. In the case where the reference layer and free layer are aligned parallel, the angular momentum transfer works in the opposite fashion, such that the formation of the ferromagnetic state, and therefore switching itself, is hindered. This results in a breaking of the symmetry of toggle switching, and could provide a regime where switching only occurs from an antiparallel to a parallel state, and not vice versa. In passing we note that due to the relatively high total magnetic moment of the reference layer in our samples compared to previous work[22], we do not expect it to be significantly influenced by any spin current originating from the free layer. Moreover, in our work the volume of magnetized Gd, which was conjectured to govern this 'reverse' spin current, is relatively small, yielding a small spin current in any case. After a few picoseconds, the reference layer should therefore start to remagnetize to its original saturation magnetization, ensuring that it remains fixed.

To experimentally confirm the expected behaviour of our system, we investigate the switching behaviour after excitation by single laser pulses with a duration of ~100 fs in a sample which is prepared in either the P or AP state. In Fig. 1c, we present the results of these measurements, where we determine the threshold laser fluence $F_0$ needed for switching by extracting the size of a switched domain as a function of incident laser pulse energy[13] (see 'Methods'). We find that the threshold fluences for switching are indeed not the same for both states. The fluence needed to switch from the AP state ($F_{0,D}$) is 1 mJ/cm² lower than the fluence needed to switch from the P state ($F_{0,T}$). Similar data for all four possible states are shown in Supplementary Fig. 1.

The difference in switching fluence is in accordance with the expectation that the spin current generated in the reference layer either assists or hinders switching from the AP or P state, respectively. As a consequence, a regime of laser fluences (indicated by the green region between the fits in Fig. 1c) now

exists where part of an excited region will only switch from an AP state to the corresponding P state, and never back with the same fluence. In other words, in this regime the magnetization is deterministically written to the P state. For higher fluences, indicated as the blue region in Fig. 1c, switching back from a P state is possible and toggle switching is recovered. Note that the laser spot radius is kept constant for both measurements, so the asymmetry between the states can also directly be seen in the different threshold energy for deterministic writing ($E_D$) and toggle switching ($E_T$).

By making use of both this difference in threshold fluence and the Gaussian spatial energy density profile of the laser pulse, we can demonstrate both deterministic writing and toggle switching in a single experiment, as shown in Fig. 1d. As sketched in the top part of this figure, both mechanisms should be present across a single laser pulse with high enough maximum fluence, assuming the switching is governed by local energy dissipation. In the bottom part of Fig. 1d, we present microscope images with magnetization contrast (Kerr microscopy, see 'Methods' and Supplementary Note 4) of a sample where both an AP and a P region are excited by a laser pulse with such a high maximum fluence. Here it can now be directly seen that the size of the domain written by a single laser pulse indeed depends on the initial magnetization state, as seen previously in Fig. 1c. This is contrary to the 'standard' toggle AOS behaviour, where there is no such symmetry breaking for the energy needed to induce a switch. The demonstration of deterministic magnetization writing can be seen in the Kerr images by the larger area which is switched only when the magnetization starts in the AP state. This outer region does not switch when starting from the P state due to hindering by the spin current, which is aligned with the free layer in that case. Note that for lower total laser pulse energies, no switching will occur from the P state and an entire domain will be written deterministically, as we will demonstrate in the following.

**Writing both states of the free layer deterministically.** As a full demonstration of the ability to deterministically write both states of the free layer, we present a scheme taking advantage of the switching behaviour in these samples in Fig. 2a. Two well-defined procedures can be used to write the desired state of the free layer relative to the reference layer (P or AP). By using a single laser pulse with maximum fluence above the threshold fluence for deterministic writing $F_{0,D}$ but below the toggle switching threshold $F_{0,T}$, only a P state can be written. A second procedure is used to write the corresponding AP state. In this case, a single pulse with maximum fluence $F_{0,D} < F_{pulse} < F_{0,T}$ first ensures the magnetization is in the P state. Subsequently, the same region is exposed to a second pulse with maximum fluence $F_{pulse} > F_{0,T}$. The part of the pulse that meets this condition will then be able to switch the P region to AP. In Fig. 2b we experimentally demonstrate this scheme with Kerr microscopy images of samples exposed in the previously described manner. It can be seen that in the centres of all exposed areas (indicated by the black squares), the final state is indeed fully dependent on the writing procedure, and no longer on the initial magnetization state. Additionally, we observe deterministic writing in the outer regions for the two-pulse procedure, due to the higher total energy of the second pulse. Nevertheless, the existence of any region where deterministic writing of both free layer states is possible is sufficient for applications using patterned media or by making use of plasmonic structures to provide local heating[31]. Although the use of two pulses might somewhat limit the speed of this protocol, it has been shown theoretically that a second laser pulse can be used to switch the magnetization a second time before the system has fully relaxed[32]. In this case the delay between the two pulses could

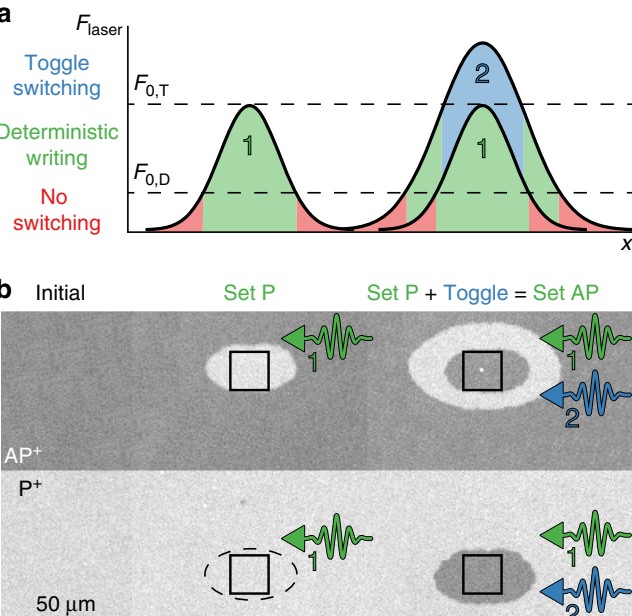

**Fig. 2 Controllably writing both states of the free layer. a** Sketch of the scheme used to set the free layer magnetization parallel to the reference layer with one fs laser pulse (left) or antiparallel with two pulses (right). **b** Kerr microscopy images showing the deterministic writing of both a P and AP state (black squares) regardless of the initial state of the switching layer.

be on the order of a few ps. Note that more experiments on the systems described in this work should be performed to determine the relevant timescales.

**Tuning the spin current magnitude.** To verify that the origin of the symmetry breaking is truly non-local transfer of angular momentum, we investigate the impact of the magnitude of the generated spin current on the regime where deterministic writing is possible. Note that a more straightforward, though less quantitative verification is presented in Supplementary Note 3. By tuning the Ni thickness in the reference layer, we change the total magnetic moment and consequently the magnitude of the spin current[30]. Here, a larger volume of magnetic material should result in a stronger spin current.

The thickness of each Ni layer ($t_{Ni}$) in the reference layer is varied from 0.5 to 1.0 nm, while keeping the amount of repetitions constant. We determine the threshold fluence for both deterministic writing and toggle switching ($F_{0,D}$ and $F_{0,T}$, respectively) for a range of $t_{Ni}$, using the same method as in Fig. 1c. Note that instead of preparing the sample in the AP or P state initially, this experiment is performed by exposing a sample prepared in the AP state to two identical subsequent laser pulses. This results in a ring-like domain as seen in the inset of Fig. 3a, as the first pulse will create a P domain, whereas the second will create a smaller AP domain within the first domain due to the higher threshold fluence $F_{0,T}$. This allows us to determine both sets of domain sizes from a single experiment, and reduces the variance between measurements. The extracted threshold fluences are presented in Fig. 3a. Here we observe two effects. Firstly, both sets of threshold fluences increase with increasing $t_{Ni}$. This behaviour can be explained by a decrease of optical absorption in the free layer with increasing reference layer thickness (see Supplementary Note 2), as well as a probable increase in free layer roughness. More importantly, we observe that the gap between $F_0$,

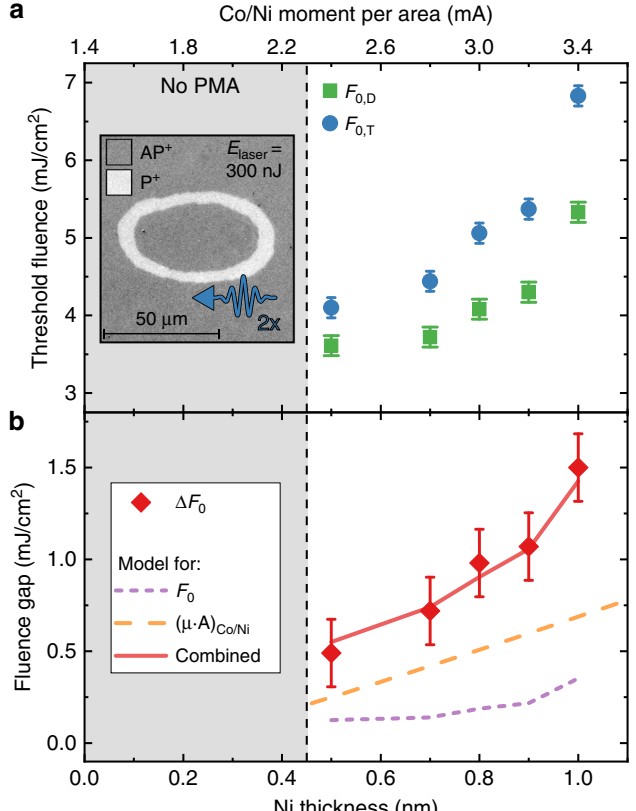

**Fig. 3 Effect of spin current magnitude on deterministic writing regime. a** The threshold fluence for switching from the AP state ($F_{0,D}$) and the P state ($F_{0,T}$) as a function of the Ni thickness $t_{Ni}$ in the $(Co(0.2)/Ni(t_{Ni}))_{x4}/Co(0.2)$ reference layer. The grey area indicates where $t_{Ni}$ was too low to obtain PMA. Inset shows a Kerr microscopy image of a typical experiment used to obtain the threshold fluences, where a sample prepared in the AP state is exposed to two subsequent identical laser pulses, yielding both domain sizes. **b** The difference $\Delta F_0$ between the fluences presented in (**a**). Lines indicate scaling behaviour with either these fluences, reference layer magnetic moment and optical absorption, or both combined. Error bars represent the standard deviation obtained through fitting.

$_D$ and $F_{0,T}$ also increases with $t_{Ni}$. This gap, defined as $\Delta F_0 = F_{0,T} - F_{0,D}$, is plotted in Fig. 3b. The increase of $\Delta F_0$ is consistent with the hypothesis that the effect is driven by a spin current originating from the reference layer.

To verify the hypothesis more quantitatively, we model the trends for increasing $t_{Ni}$. To show that the increase in $\Delta F_0$ is not merely a consequence of scaling with the increase of both threshold fluences, we present this scaling as the purple dashed curve in Fig. 3b. As it is clear that this behaviour alone cannot explain the increase in $\Delta F_0$, we now turn to the expected scaling of the spin current. The optically generated spin current should scale primarily with two factors, namely the magnetic moment of the reference layer ($\mu_{Co/Ni}$) and the optical absorption $A$. We model $\mu_{Co/Ni}$ as a function of $t_{Ni}$ by assuming bulk values for the magnetization for each sublayer. For all proposed mechanisms of optical spin current generation, the spin current scales with the light absorption in either the magnetic layer (due to demagnetization[24,33]) or all metallic layers underneath (due to hot electron generation[23]). As the absorption in all layers should scale in the same fashion with increasing thickness, we use the absorption in the reference layer itself for the sake of simplicity. Moreover, it has been found in similar Co/Ni multilayers that the

spin current originates from the full thickness of the generating layer[30]. We calculate the light absorption in the reference layer ($A_{Co/Ni}$) as a function of $t_{Ni}$ by using a transfer matrix method[34,35] (see Supplementary Note 2). The trend for both $\mu_{Co/Ni}$ and $A_{Co/Ni}$ is shown by the orange dashed curve in Fig. 3b. By now combining this with the scaling of $F_0$ discussed earlier, we find the red curve in this figure, which can be seen to adequately explain the increase of $\Delta F_0$. This further confirms that the corresponding symmetry breaking is indeed the result of an optically generated spin current. Some effects are not included here, such as the dependence of the reference layer demagnetization rate on the multilayer composition. Indeed, different demagnetization dynamics could arise at high fluences for different Ni concentrations[36] or due to variation in the Curie temperature[37]. Moreover, it should be noted that the actual magnetization of the reference layer likely differs from the values assumed here due to an absence of perfectly sharp multilayer interfaces. However, assuming that these effects are small compared to the variation in absorption and magnetization we have shown that this relatively simple description can adequately explain the data. Finally, we would like to draw attention to the magnitude of the effect. For a $(Co(0.2)/Ni(1))_{x4}/Co(0.2)$ reference layer the threshold fluence gap $\Delta F_0$ has a relatively large magnitude, corresponding to 28% of the base fluence $F_{0,D}$. This is expected to be scaleable towards even larger values, for instance by using stack engineering to increase the magnetization of the reference layer and light absorption in this layer.

## Discussion

In conclusion, we have experimentally demonstrated deterministic writing of the magnetization of a free Co/Gd bilayer using an fs laser pulse by making use of the symmetry breaking provided by a spin current generated in a neighbouring ferromagnetic Co/Ni reference layer. Moreover, we have demonstrated two protocols for reliably and controllably writing both (bit) states of the free layer, by using either one or two laser pulses. We have also shown that the spin current induced symmetry breaking scales as expected with the magnitude of the spin current generated in the reference layer, by tuning its composition. The system described here benefits from the strong binary threshold of AOS and can provide a versatile method to further enhance the general understanding of optically generated spin currents. By for instance changing the time delay between the arrival of the spin current and the switching of the free layer, the resulting switching behaviour could be used to map the temporal profile and magnitude of the spin current. Moreover, this system can provide insight into the role of spin transport versus local transfer of angular momentum in AOS, which is essential for the implementation of future opto-spintronic devices, such as the optically written racetrack memory[25,38,39]. The deterministic magnetization writing presented in this work provides an important stepping stone on the road to realizing such data storage devices.

## Methods

**Sample fabrication**. The samples in this work were prepared via DC magnetron sputtering, with a base pressure in the deposition chamber of ~$10^{-9}$ mbar. Si wafers coated with a 100-nm $SiO_2$ layer were used as substrate, as this layer acts as a reflection coating and enhances optical absorption. The general sample structure for all measurements is $Ta(4)/Pt(4)/(Co(0.2)/Ni(t_{Ni}))_{x4}/Co(0.2)/Cu(5)/Pt(0.5)/Co(1)/Gd(3)/Ta(4)$, where the numbers between parentheses indicate layer thicknesses in nm. Ta is used as seeding layer for Pt to ensure the proper (111) texture, and as a capping layer. The Pt layers induce PMA in both the reference layer and free layer.

**Measurements**. The hysteresis loop of the sample was measured in a static polar MOKE setup.
   AOS was performed with linearly polarized laser pulses with a pulse width of ~100 fs at sample position, and a central wavelength of 700 nm. By using a pulse

picker and a mechanical shutter, individual pulses could be selected to excite the sample. In order to determine the threshold fluence the samples were excited at different locations with increasing laser pulse energy.

Subsequently, Kerr microscopy images of the excited regions were obtained using a differential technique to enhance magnetic contrast.

These images were analysed using standard image analysis routines to obtain the written domain size as a function of laser pulse energy. These data were fitted assuming a Gaussian energy profile of the laser pulse, giving the threshold fluence for switching[13,40].

All measurements in this work were performed at room temperature.

## Data availability
The data that support the findings of this study are available from the corresponding author on reasonable request.

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

## Acknowledgements
This work is part of the Gravitation programme 'Research Centre for Integrated Nanophotonics', which is financed by the Netherlands Organisation for Scientific Research (NWO).

## Author contributions
Y.L.W.v.H. fabricated the structures and designed the experiments. Y.L.W.v.H. and P.v.d.M. conducted the experiments and analysed the data. B.K. and R.L. supervised the project. Y.L.W.v.H. wrote the manuscript with input from all authors.

## Competing interests
The authors declare no competing interests.
