## [Peer Review File · Nature Communications]

Reviewers' Comments:

Reviewer #1:

Remarks to the Author:

The manuscript by van Hees et al. aims at demonstrating deterministic all-optical magnetic switching for a material, Co/Gd bilayers, where previously mainly so-called toggle switching (magnetization reversal for each fs laser pulse) was observed. They achieve this by combining toggle switching with spin currents induced by demagnetizing an adjacent CoNi multilayer. The results are new and provide interesting insight into the interplay of the two mentioned effects, e.g. by varying the spin current strength. The proposed pulse sequences also provide an appealing outlook for applications. I would therefore support publication in Nature Communications after several inconsistencies are removed:

1. It is mentioned several times in the manuscript that all-optical switching can be "defined by the helicity of many incoming laser pulses, or toggled by a single pulse" (abstract). The former is not generally true, see PRL 103, 117201 (2009). This reference also contradicts the authors' subsequent sentence in the abstract.
2. I find it confusing that the authors call their method 'single pulse deterministic magnetization writing' (p. 3 last paragraph) but then admit that up to 2 pulses are needed for writing. Surely they can find a better description.
3. A similar experiment (ref. [20]) generating different/complementary results is briefly mentioned twice. Since the present paper can be considered a follow up experiment a thorough discussion of the present results in light of ref. [20] is required. For instance, why does in one case the reference layer switch and in the other case it does not? What are the relative strengths of spin currents from and to the free layer? Does the nature of the reference layer or the free layer, e.g. alloy vs. bilayer etc., play a role?
4. In Fig. 3a the greyscales for AP+ and P+ states seem to be somewhat off. At least the legend does not match the figure. In this respect: How do the authors differentiate their proposed switching from that in ref. [20]? How can they differentiate in the microscopy images switching P+ to AP+ from, say, P+ to AP-? I understand MOKE hysteresis loops were measured in a different setup. What is the accuracy of doing this in the microscope? A figure or better description would help.

Reviewer #2:

Remarks to the Author:

Toggle switching of GdFeCo alloys by single laser pulses has attracted a lot of interest lately since it bears the potential to speed up applications based on magnetic switching orders of magnitude with respect to traditional switching of magnetic devices. Previous works have demonstrated Gd/Co bilayers to also show the same toggle switching.

For magnetic recording purposes toggle switching is inconvenient and one would prefer a scheme where at each instance one could decide to which direction the polarity should switch.

The authors address this question by proposing a scheme based on adding a ferromagnetic reference layer to Gd/Co bilayers. The magnetic state of the reference breaks the symmetry of the total system under switching of Gd/Co magnetic polarity.

The authors conduct the necessary experiments to demonstrate their idea. They first demonstrate that there is a laser fluence window where deterministic switching is possible. Second, they use the fact that toggle switching and deterministic switching need of a different laser fluence to propose a protocol for magnetic recording. When two consecutive pulses with laser fluence necessary for i) deterministic switching and ii) toggle switching are applied, a well localized region of the sample can be switched at will. This protocol and its demonstration is new.

Perhaps, the main shortcoming of this protocol is the necessity to use two pulses, which have to

be separated by some time delay, probably around nanoseconds, to recover the equilibrium initial state. This would slow down the magnetic writing effectively to the nanosecond time scale. The authors might consider discussing the possibility to speed up the proposed protocol by applying two shortly-delayed pulses to their structure. The following works on the effect of applying two consecutive shortly-delayed laser pulses could be of interest:

* 'Interplay of heating and helicity in all-optical magnetization switching

Sabine Alebrand, Alexander Hassdenteufel, Daniel Steil, Mirko Cinchetti, and Martin Aeschlimann, Phys. Rev. B 85, 092401 (2012)

* Ultrafast double magnetization switching in GdFeCo with two picosecond-delayed femtosecond pump pulses

Appl. Phys. Lett. 113, 062402 (2018)

The physics behind this symmetry breaking is related to the supposed spin currents coming from the reference layer and acting somehow on the Gd/Co magnetization of the free layer during the switching process. These physics is not demonstrated in this paper but it can be inferred that this could be the case from the data shown in Figure 3. I believe the possibility of controlling AOS with ultrafast spin currents is a great idea and this work could be a stepping stone along this line. However, I feel that a deeper discussion is necessary about the physics of the spin currents that could be optically generated in the reference layer, and how it travels and it is injected into Co/Gd, how exactly the scattering between mobile and localized electron occurs. Would be more efficient to keep the reference layer magnetization transversal to the free layer magnetization to obtain the maximum torque from the scattering of mobile electron's spin and the local Co spins?

To summarize, the work presented in the current manuscript is excellent, well-conducted, the manuscript is well-written, the idea is new, the protocol for magnetic recording works, and the physics behind this proposal seems to make sense and the authors have somehow (not completely) demonstrated that the non-local transfer of spin angular momentum could be the origin of observed behavior. I would therefore like to see it published.

Unai Atxitia

Reviewer #3:

Remarks to the Author:

The manuscript by Y. Hees et al discusses the novel possibility to use laser-induced spin current pulses for assisting the switching of magnetization. Laser irradiation of a sample consisting of a free and reference layers results in the generation of perpendicular spin current operating on the femtosecond timescale. The authors analyze various regimes of the spin current-assisted switching and conclude that depending on the mutual alignment of the magnetizations in the free and reference layers, spin current pulse can facilitate or hinder switching of magnetization in the free layer. Based on these findings, they discuss switching protocols allowing deterministic magnetization switching. The manuscript is well organized and clearly written.

The experimental results presented in the manuscript are very interesting and definitely are worth publishing in one way or another. However, in my view the manuscript has a very serious leaning towards applied research and does little to address fundamental questions pertinent to magnetization switching. For instance, in the conclusions section the only point relevant for the fundamental issues is "the system described here [...] can provide a versatile method to further enhance the general understanding of optically generated spin currents". This is, in my opinion, not enough to justify the publication just yet. The manuscript could be significantly improved by further analysis of the experimental results, in particular, in what concerns the optical generation of spin current pulses in such complex multilayer systems as the one used in this work. Should the

authors imply that deterministic switching of magnetization can be used for probing spin polarization of laser-induced current pulses, this could be discussed more closely.

Furthermore, the optical generation of spin current in the Co/Ni multilayer deserves further attention. The manuscript would strongly benefit from an analysis of this process, especially including the rather complex excitation scheme (Fig. 1). Here, one would expect two competing spin current pulses propagating in the opposite directions. Moreover, what is the mechanism of the spin current generation? If it is similar to the one discussed in Ref. 25, how does the increased number of interfaces influence the spin polarization degree? How well-defined are the interfaces and if there is "additional alloying during growth" (page 11), what is its role with regards to the spin scattering at interfaces? What is the characteristic depth from which the spin current is effectively generated? All these are important questions which should be discussed in the manuscript.

We would like to thank all reviewers for the time invested in reviewing our manuscript and for offering constructive feedback. We believe that the changes made as a result of their comments have improved the quality of the manuscript, mainly regarding some questions shared by all reviewers about the underlying physics. In the following we address the comments of all reviewers (red cursive text) individually and mark any changes to the manuscript in blue cursive text.

Reviewer #1:

The manuscript by van Hees et al. aims at demonstrating deterministic all-optical magnetic switching for a material, Co/Gd bilayers, where previously mainly so-called toggle switching (magnetization reversal for each fs laser pulse) was observed. They achieve this by combining toggle switching with spin currents induced by demagnetizing an adjacent CoNi multilayer. The results are new and provide interesting insight into the interplay of the two mentioned effects, e.g. by varying the spin current strength. The proposed pulse sequences also provide an appealing outlook for applications. I would therefore support publication in Nature Communications after several inconsistencies are removed:

We thank the reviewer for the positive feedback regarding our manuscript. In the following we address the reviewer's concerns.

1. It is mentioned several times in the manuscript that all-optical switching can be "defined by the helicity of many incoming laser pulses, or toggled by a single pulse" (abstract). The former is not generally true, see PRL 103, 117201 (2009). This reference also contradicts the authors' subsequent sentence in the abstract.

We agree with this comment. It has indeed been shown in ferrimagnetic GdFeCo alloys that the magnetization can be switched with a single circularly polarized laser pulse, and can only be switched back with a laser pulse with the opposite helicity. This has been shown to be the result of a dependence of the laser absorption on the magnetization state, i.e. magnetic circular dichroism (MCD). As this is a small effect, this behavior only manifests in a relatively narrow range of laser fluences. For the sake of clarity and completeness we have amended the relevant sentences as follows.

Abstract:

Two distinct mechanisms have been observed, where the final magnetization state is either defined by the helicity of several incoming laser pulses in many different magnetic materials, or switched by a single pulse in certain ferrimagnets. Although the latter is highly efficient and ultrafast, in most cases it is a toggle mechanism, which is not desirable for applications. In this work we experimentally demonstrate conversion from toggle switching to a deterministic mechanism...

Second paragraph:

It should be noted that single pulse helicity dependent switching has also been observed in a narrow range of laser fluences for GdFeCo alloys [2,15], owing to magnetic circular dichroism, i.e. magnetization- and helicity-dependent light absorption [16].

To support this we have included two new references (numbers as in new version of manuscript):

[15] Vahaplar et al. Phys. Rev. Lett. 103, 117201 (2009)

[16] Khorsand et al. Phys. Rev. Lett. 108, 127205 (2012)

2. I find it confusing that the authors call their method ‘single pulse deterministic magnetization writing’ (p. 3 last paragraph) but then admit that up to 2 pulses are needed for writing. Surely they can find a better description.

Although switching from the AP to the P state is a single pulse mechanism, the protocol we propose for switching to the AP state indeed needs a sequence of two pulses. We therefore agree with the reviewer that this choice of words is not optimal, and have changed the offending phrase to “*deterministic magnetization writing using either one or two laser pulses*”. Moreover, we have emphasized in the relevant locations that two pulses are needed for full writing capabilities.

3. A similar experiment (ref. [20]) generating different/complementary results is briefly mentioned twice. Since the present paper can be considered a follow up experiment a thorough discussion of the present results in light of ref. [20] is required.

Although we feel that within the scope of this work the key differences with the work of Iihama et al. have been sufficiently discussed on page 7, we will address a few of the specific questions in more detail below.

For instance, why does in one case the reference layer switch and in the other case it does not? Does the nature of the reference layer or the free layer, e.g. alloy vs. bilayer etc., play a role?

In the case of the system of Iihama et al. the main contribution of the spin current was found to be the Gd sublattice. In their case, a GdFeCo alloy of 5 nm thick containing ~22% Gd was used. In our systems, the amount of magnetized Gd near the Co/Gd interface was found to correspond to approximately 0.45 nm of fully magnetized Gd. Therefore any possible spin current coming from the Gd would be at most half as strong in our system. Moreover, this spin current has to pass through the Co layer before being able to reach the reference layer, which would also weaken it significantly.

The Co/Ni multilayer in our system also consists of more magnetic material by volume compared to the system of Iihama et al. Our reference layer will be more resilient against the influence of an external spin current compared to the Co/Pt multilayer.

This finally leaves a possible spin current coming from the Co layer in the free Co/Gd bilayer, which leads to the next point:

What are the relative strengths of spin currents from and to the free layer?

If we assume (as we do in the main text) that the spin current scales with the magnetization of the layer from which it originates (as supported by Laliou et al. [1]), we can estimate that the spin current coming from the Co layer is 2.2 to 1.3 times weaker than the spin current coming from the Co/Ni reference layer (depending on Ni thickness). However, we believe that the ability of the reference layer spin current to influence the switching is intrinsically linked to the non-equilibrium state of the free layer during switching. As we mention in the main text, it has been shown that such a spin current significantly influences the demagnetization of a ferromagnetic layer [2]. It is therefore not necessarily a matter of relative strengths of the spin current, but rather the clear threshold nature of single pulse AOS that allows us to observe this effect very clearly.

This also highlights again why the reference layer does not switch. At the fluences where switching of the Co/Gd bilayer occurs, the Co/Ni multilayer does not end up in a multi-domain state, meaning it has not been fully demagnetized. Due to the remaining finite magnetic moment a spin current coming from the free layer has a less pronounced effect on the reference layer than vice versa.

In response to both this reviewer and reviewer #3 we have addressed this in the manuscript, in the section “**Deterministic optical magnetization writing**”:

We note that a spin current is also generated in the free layer, travelling towards the reference layer. However, as this particular spin current is much weaker due to the reduced magnetic moment of the free layer and is not found to significantly influence the receiving reference layer, we have not sketched it here. Moreover, it is precisely the threshold nature of AOS that allows us to observe the effect of a spin current much more clearly than in the reference layer, which is only partially demagnetized.

4. In Fig. 3a the greyscales for AP+ and P+ states seem to be somewhat off. At least the legend does not match the figure.

The legend in this figure has erroneously been adapted from Fig. 1b, where the contrast settings of the Kerr microscope were slightly different. We thank the reviewer for noticing this error, and have amended it.

In this respect: How do the authors differentiate their proposed switching from that in ref. [20]? How can they differentiate in the microscopy images switching P+ to AP+ from, say, P+ to AP-? I understand MOKE hysteresis loops were measured in a different setup. What is the accuracy of doing this in the microscope? A figure or better description would help.

In principle hysteresis loops can indeed be measured in the Kerr microscope, in which case the contrast levels can be directly matched to the magnetization states. However, we have used a different method to perform this matching. We have exposed the samples to a pulse train for around a minute, yielding thermal demagnetization of both the reference and free layer. Due to the overlapping of these domains all four possible configurations are present in the irradiated region, allowing us to match them to the contrast levels. This method is detailed further in the newly added Supplementary Note 4.

Moreover, this method gives more insight regarding this reviewer’s question. We observe that when the sample is initially in a plus (+) state, the formation of a minus (-) state (meaning a switch of the reference layer) only occurs upon either prolonged laser heating of the sample, or exposure with a single pulse with a very high energy. This shows that the reference layer merely demagnetizes at laser fluences where the free layer, again highlighting that the reference layer is much more stable than the free layer.

[1] M. L. M. Laliu, P. L. Helgers, and B. Koopmans, “Absorption and generation of femtosecond laser-pulse excited spin currents in noncollinear magnetic bilayers,” Physical Review B 96, 014417 (2017).

[2] G. Malinowski et al., “Control of speed and efficiency of ultrafast demagnetization by direct transfer of spin angular momentum,” Nature Physics 4, 855 (2008).

Reviewer #2:

Toggle switching of GdFeCo alloys by single laser pulses has attracted a lot of interest lately since it bears the potential to speed up applications based on magnetic switching orders of magnitude with respect to traditional switching of magnetic devices. Previous works have demonstrated Gd/Co bilayers to also show the same toggle switching.

For magnetic recording purposes toggle switching is inconvenient and one would prefer a scheme where at each instance one could decide to which direction the polarity should switch.

The authors address this question by proposing a scheme based on adding a ferromagnetic reference layer to Gd/Co bilayers. The magnetic state of the reference breaks the symmetry of the total system under switching of Gd/Co magnetic polarity.

The authors conduct the necessary experiments to demonstrate their idea. They first demonstrate that there is a laser fluence window where deterministic switching is possible. Second, they use the fact that toggle switching and deterministic switching need of a different laser fluence to propose a protocol for magnetic recording. When two consecutive pulses with laser fluence necessary for i) deterministic switching and ii) toggle switching are applied, a well localized region of the sample can be switched at will. This protocol and its demonstration in new.

We thank the reviewer for the thorough analysis of our work.

Perhaps, the main shortcoming of this protocol is the necessity to use two pulses, which have to be separated by some time delay, probably around nanoseconds, to recover the equilibrium initial state. This would slow down the magnetic writing effectively to the nanosecond time scale. The authors might consider discussing the possibility to speed up the propose protocol by applying two shortly-delayed pulses to their structure. The following works on the effect of applying two consecutive shortly-delayed laser pulses could be of interest:

* *Interplay of heating and helicity in all-optical magnetization switching*

Sabine Alebrand, Alexander Hassdenteufel, Daniel Steil, Mirko Cinchetti, and Martin Aeschlimann, Phys. Rev. B 85, 092401 (2012)

* *Ultrafast double magnetization switching in GdFeCo with two picosecond-delayed femtosecond pump pulses*

Appl. Phys. Lett. 113, 062402 (2018)

We agree with the reviewer that this factor should have been mentioned in the manuscript. In the past year we have already been inspired by the second reference proposed by the reviewer. We are currently preparing a manuscript regarding experimental work on the minimum time delay with which a second pulse can switch the magnetization again in a simple Co/Gd bilayer system. The insights obtained there could be applied to the system under investigation in this work.

We have amended the manuscript to address the concern of the delay between the two pulses to the main text (section **Writing both states of the free layer deterministically**):

Although the use of two pulses might somewhat limit the speed of this protocol, it has been shown theoretically and experimentally that a second laser pulse can be used to switch the magnetization a second time before the system has fully relaxed [32,33]. In this case the delay between the two pulses is on the order of a few tens of ps. Note that these works only concern 'simple' GdFeCo or Co/Gd systems, and more experiments on the systems described in this work should be performed to determine the relevant timescales.

The added references are (numbers as in new version of manuscript):

[32] U. Atxitia and T. Ostler, Applied Physics Letters 113, 062402 (2018)

[33] Y.L.W. van Hees, B. Koopmans, and R. Lavrijsen, In preparation

Moreover, we would like to show preliminary results of Ref. 33 here to convince the reviewer that switching back within some tens of ps is indeed experimentally possible in Co/Gd.

Fig. R1: Experiment to determine the timescale at which a second fs laser pulse can switch the magnetization again in a Co/Gd bilayer sample. Pulse 1 and 2 are made to spatially overlap on the sample and excite the sample with a time delay of 50 ps. The green arrow indicates a region which is fully switched twice [33].

In Fig. R1 we show a Kerr microscopy image of a Co/Gd sample that has been exposed to two fs laser pulses with a time delay of 50 ps. By examining the region which has been affected by both pulses we find that the outer edge of this region (as indicated by the green arrow) shows a tendency to switch back to the initial state. Although a multi-domain state is visible, there is a clear preference for switching at these timescales. This effect rapidly becomes less clear when decreasing the time delay, but remains visible down to a time delay of 20 ps. By further engineering the sample structure and carefully tuning the laser fluence we will further examine what governs this ‘back-switching’ in [33].

In principle we see no reason that this behavior should not apply to the more complex systems discussed in the work under review. This would bring the time needed to write an AP state using 2 pulses much closer to that of the single pulse writing procedure, as in both cases the system should be allowed to relax for some tens of ps before another writing sequence can be performed. However, while we are planning to repeat the experiment shown here on the structures presented in the manuscript, this is currently outside of the scope of this work.

The physics behind this symmetry breaking is related to the supposed spin currents coming from the reference layer and acting somehow on the Gd/Co magnetization of the free layer during the switching process. These physics is not demonstrated in this paper but it can be inferred that this could be the case from the data shown in Figure 3. I believe the possibility of controlling AOS with ultrafast spin currents is a great idea and this work could be a stepping stone along this line.

We are pleased to hear that the reviewer agrees with our vision of combining AOS with ultrafast optically generated spin currents. It is our belief that this is indeed an elegant way to exert more control over the AOS process, which would be very beneficial for future applications.

However, I feel that a deeper discussion is necessary about the physics of the spin currents that could be optically generated in the reference layer, and how it travels and it is injected into Co/Gd, how exactly the scattering between mobile and localized electron occurs. Would be more efficient to keep the reference layer magnetization transversal to the free layer magnetization to obtain the maximum torque from the scattering of mobile electron's spin and the local Co spins?

From previous experience with using Co/Ni multilayers to optically generate spin currents [1] we know that in these systems (i.e. in the thin limit) the spin current scales approximately linearly with the thickness of the reference layer. This implies that the entire reference layer contributes to the generation of the spin current for the thickness range used here. With the current experimental results we can only speculate as to the physical origin of the spin current in these particular systems, but note that we already mention the possible candidates that have been proposed (dM/dt-like, superdiffusive, and spin-dependent Seebeck effect) in the main manuscript. As highlighted in [1], the behavior of the spin current with increasing thickness of the generating layer seems at odds with the behavior of a superdiffusive spin current. Moreover, any contribution from the spin-dependent Seebeck effect has been shown to be negligible in ferromagnetic multilayers such as this [3]. However, without stronger experimental evidence we do not wish to posit one generation mechanism in our structures in this particular work.

As for the absorption in the free layer, we believe that spin flip scattering, rather than the exertion of a torque by the mobile electrons on the local spins is the driving mechanism behind the observed effect. In contrast to the work presented in [1,3], in our work the moment of the Co layer is strongly quenched as a necessity for AOS, which severely limits the existence of a sizable torque. We do agree that it would be interesting to investigate the effect of a transversal spin current on switching in these systems, however this is outside of the scope of this work. Moreover, while this would not break the symmetry of the switching preference, it could however with some speculation lead to a strongly reduced threshold fluence.

To summarize, the work presented in the current manuscript is excellent, well-conducted, the manuscript is well-written, the idea is new, the protocol for magnetic recording works, and the physics behind this proposal seems to make sense and the authors have somehow (not completely) demonstrated that the non-local transfer of spin angular momentum could be the origin of observed behavior. I would therefore like to see it published.

We thank the reviewer for the positive comments regarding our work.

[1] M. L. M. Laliou, P. L. Helgers, and B. Koopmans, "Absorption and generation of femtosecond laser-pulse excited spin currents in noncollinear magnetic bilayers," Physical Review B 96, 014417 (2017).

[3] A.J. Schellekens et al. "Ultrafast spin-transfer torque driven by femtosecond pulsed-laser excitation." Nature Communications 5: 4333 (2014).

Reviewer #3:

The manuscript by Y. Hees et al discusses the novel possibility to use laser-induced spin current pulses for assisting the switching of magnetization. Laser irradiation of a sample consisting of a free and reference layers results in the generation of perpendicular spin current operating on the femtosecond timescale. The authors analyze various regimes of the spin current-assisted switching and conclude that depending on the mutual alignment of the magnetizations in the free and reference layers, spin current pulse can facilitate or hinder switching of magnetization in the free layer. Based on these findings, they discuss switching protocols allowing deterministic magnetization switching. The manuscript is well organized and clearly written.

We appreciate the analysis of and positive comments regarding our work.

The experimental results presented in the manuscript are very interesting and definitely are worth publishing in one way or another. However, in my view the manuscript has a very serious leaning towards applied research and does little to address fundamental questions pertinent to magnetization switching.

In our letter we propose and experimentally demonstrate conceptually an entirely novel scheme of deterministic switching. Moreover, we do not claim that this specific work directly provides answers to these fundamental questions, but rather that it provides a stepping stone in this regard. In the following we address the specific questions posed by this reviewer on this matter.

For instance, in the conclusions section the only point relevant for the fundamental issues is “the system described here [...] can provide a versatile method to further enhance the general understanding of optically generated spin currents”. This is, in my opinion, not enough to justify the publication just yet. The manuscript could be significantly improved by further analysis of the experimental results, in particular, in what concerns the optical generation of spin current pulses in such complex multilayer systems as the one used in this work. Should the authors imply that deterministic switching of magnetization can be used for probing spin polarization of laser-induced current pulses, this could be discussed more closely.

This is indeed a possible application of these systems on which we have not been very clear. We expand on a specific possibility in the conclusion as follows:

By for instance changing the time delay between the arrival of the spin current and the switching of the free layer, the resulting switching behaviour could be used to map the temporal profile and magnitude of the spin current.

We envision performing such an experiment by optically decoupling the reference and free layer and pumping both layers separately with a variable time delay. As we show the effect to be quite sizable already, we expect a temporal mapping of the spin current to be realistic. In doing so we could compare this temporal profile with the expected profiles for a dM/dt -like or a superdiffusive spin current, and provide more insight into the generation mechanism in these ferromagnetic multilayer stacks.

Furthermore, the optical generation of spin current in the Co/Ni multilayer deserves further attention. The manuscript would strongly benefit from an analysis of this process, especially including

the rather complex excitation scheme (Fig. 1). Here, one would expect two competing spin current pulses propagating in the opposite directions.

We agree with the reviewer that a counterpropagating spin current coming from the Co/Gd free bilayer should also exist. We have chosen not to sketch it in this figure for the sake of clarity. In response to remark 3 by reviewer #1 we have highlighted why this spin current should be weaker and its effect much smaller than that coming from the reference layer. For clarity we mention again that these questions have been addressed in the manuscript, in the section **“Deterministic optical magnetization writing”**:

We note that a spin current is also generated in the free layer, travelling towards the reference layer. However, as this particular spin current is much weaker due to the reduced magnetic moment of the free layer and is not found to significantly influence the receiving reference layer, we have not sketched it here. Moreover, it is precisely the threshold nature of AOS that allows us to observe the effect of a spin current much more clearly than in the reference layer, which is only partially demagnetized.

Moreover, what is the mechanism of the spin current generation? If it is similar to the one discussed in Ref. 25, how does the increased number of interfaces influence the spin polarization degree?

For this question we reiterate our response to the final question posed by reviewer #2. We highlight all proposed mechanisms for spin current generation, as we cannot provide a conclusive answer based on our experimental results. However, it has been found that in these relatively thin multilayer systems [3] the contribution from the spin-dependent Seebeck effect should be negligible.

How well-defined are the interfaces and if there is “additional alloying during growth” (page 11), what is its role with regards to the spin scattering at interfaces?

Here we again wish to refer to the work of Lalieu et al. [1]. In this work the spin current efficiency was found to remain constant with increasing thickness of the generating layer. Increasing this thickness was done by increasing the number of repeats, and therefore the number of interfaces. If scattering at the interfaces is significant, one would expect a decreasing efficiency with thickness. We therefore assume that this factor is not of significant relevance in our systems.

We do wish to stress that we do not know the degree of alloying, and have therefore amended this hypothetical remark to:

Moreover, it should be noted that the actual magnetization of the reference layer likely differs from the values assumed here due to an absence of perfectly sharp multilayer interfaces.

What is the characteristic depth from which the spin current is effectively generated? All these are important questions which should be discussed in the manuscript.

We agree that this question, which was also posed by reviewer #2, should be addressed in the manuscript. We have done so in the section **“Tuning the spin current magnitude”**:

Moreover, it has been found in similar Co/Ni multilayers that the spin current originates from the full thickness of the generating layer.

[1] M. L. M. Lalieu, P. L. Helgers, and B. Koopmans, “Absorption and generation of femtosecond laser-pulse excited spin currents in noncollinear magnetic bilayers,” Physical Review B 96, 014417 (2017).

[3] A.J. Schellekens et al. "Ultrafast spin-transfer torque driven by femtosecond pulsed-laser excitation." *Nature Communications* 5: 4333 (2014).

Reviewers' Comments:

Reviewer #1:

Remarks to the Author:

The authors have addressed my comments satisfactorily and I recommend publication of the manuscript in its present form.

Reviewer #2:

Remarks to the Author:

I believe the authors have addressed adequately all comments from my side as well as from the other reviewers.

While the content and main point raised by this work remain unaltered, the modifications of the manuscript based on the reviewers suggestions have improved considerably its quality.

Base on the arguments already provided in my previous report, and in the view of the current version of the manuscript, I strongly support its publication in Nature Communications as it is.

Reviewer #3:

Remarks to the Author:

I have read the revised version of the manuscript entitled "Deterministic all-optical magnetization writing facilitated by non-local transfer of spin angular momentum" by van Hees et al. as well as their response to the comments of the Referees and I would like to thank the authors for improving the clarity of the paper. All issues relevant for the understanding of the contents of the paper have been clarified in the revised version. Yet, the fundamental shortcoming of the manuscript is still there: it has a rather descriptive character and lack the discussion of the spin current dynamics (and kinetics) as well as the timescales of the magnetization dynamics in the switched magnetic layer. As such, the message conveyed by van Hees et al. would be a very good fit for a Perspective or Opinion (or similar) type of article, but not the regular scientific publication. Another option, in my opinion, would be to publish it in a more applied journal where it belongs. Therefore I would like to thank the authors for their work but cannot recommend the manuscript for publication in Nature Communications.

Reviewer #1:

The authors have addressed my comments satisfactorily and I recommend publication of the manuscript in its present form.

We thank the reviewer for the time invested in reading our manuscript, and we are pleased to hear that the reviewer supports publication.

Reviewer #2:

I believe the authors have addressed adequately all comments from my side as well as from the other reviewers.

While the content and main point raised by this work remain unaltered, the modifications of the manuscript based on the reviewers suggestions have improved considerably its quality.

Base on the arguments already provided in my previous report, and in the view of the current version of the manuscript, I strongly support its publication in Nature Communications as it is.

We are pleased to hear that the revisions based on previous comments have positively affected the manuscript, and we thank the reviewer for reading and reviewing it.

Reviewer #3:

I have read the revised version of the manuscript entitled "Deterministic all-optical magnetization writing facilitated by non-local transfer of spin angular momentum" by van Hees et al. as well as their response to the comments of the Referees and I would like to thank the authors for improving the clarity of the paper. All issues relevant for the understanding of the contents of the paper have been clarified in the revised version.

We thank the reviewer for the positive words regarding the revised version of the manuscript, and concur that previous feedback has contributed significantly to its clarity.

Yet, the fundamental shortcoming of the manuscript is still there: it has a rather descriptive character and lack the discussion of the spin current dynamics (and kinetics) as well as the timescales of the magnetization dynamics in the switched magnetic layer.

Although we addressed this criticism in our previous response, we do wish to stress the fact that aside from demonstrating novel functionality, we also show that an optically generated spin current can be probed with an all-optically switchable system. Although addressing the timescales and spin current dynamics is not in the scope of this work, we do hope to follow up on this in the future by making use of this elegant probing mechanism.

As such, the message conveyed by van Hees et al. would be a very good fit for a Perspective or Opinion (or similar) type of article, but not the regular scientific publication. Another option, in my opinion, would be to publish it in a more applied journal where it belongs. Therefore I would like to thank the authors for their work but cannot recommend the manuscript for publication in Nature Communications.

We thank the reviewer again for the time invested in reading the revised version of our manuscript, and for useful feedback.

Youri van Hees, on behalf of all authors.